# The Effect of Role Conflict and Professional Autonomy on the Role Performance of Patient Safety Coordinators in Small and Medium-Sized Hospitals in Korea

**DOI:** 10.3390/ijerph19159392

**Published:** 2022-07-31

**Authors:** Kyo-Yeon Park, Kyoungrim Kang

**Affiliations:** 1College of Nursing, Pusan National University, Yangsan 50612, Korea; chloeky@naver.com; 2College of Nursing, Research Institute of Nursing Science, Pusan National University, Yangsan 50612, Korea

**Keywords:** patient safety coordinator, role performance, role conflict, professional autonomy

## Abstract

This study aimed to investigate the effects of role conflict and professional autonomy on the role performance of patient safety coordinators in small and medium-sized hospitals in Korea. The participants in this cross-sectional study were 121 patient safety coordinators in general hospitals or hospitals with more than 100–300 beds. Data were collected through an online survey for about three weeks in February 2022. The variables were role conflict, professional autonomy, and role performance. In the data analysis, we employed the *t*-test, ANOVA, correlation, and multiple regression methods. Almost all (99.2%) of the participants were nurses. The lower the role conflict and the higher the professional autonomy, the better the role performance shown. As a result of analyzing the factors affecting role performance, the regression model was found to be significant (F = 6.988, *p* < 0.001). The most influential factor in role performance was professional autonomy (β = 0.279, *p* = 0.002). In conclusion, it is thought that systematic education and legal and institutional arrangements for independent roles and work regulations are needed to strengthen patient safety coordinators’ competency in small and medium-sized hospitals in Korea. This will improve the role performance of patient safety coordinators and create a better patient safety culture.

## 1. Introduction

Patient safety means the prevention of injuries or accidents that occur during the delivery of medical services [1]; as the public’s demand for medical services increases, social interest in patient safety is also increasing [2,3]. However, the Organization for Economic Cooperation and Development (OECD) pointed out that there is no clear system that guarantees patient safety, which is a concern for Korea’s health care system [4]. Since then, national-led patient safety strategies such as legislative activities related to patient safety have been established, but Korea is about 10 years behind major countries such as the USA, UK, and Japan [5].

In 2010, a child who was being treated for leukemia in Korea died due to an incorrect injection of an anticancer drug (Vincristine), and another adult patient died in the same way. Since then, accidents involving patient safety have occurred in both children and adults, and the need for a national reporting and learning system for patient safety accidents has been emphasized. The Patient Safety Act was enacted in 2015 [6]. Prior to this, there was no systematic patient safety system at the national level, so accurate statistics on patient safety accidents were not known [7]. However, since the Patient Safety Act was brought into force in 2016, patient safety coordinators have been allocated to most medical institutions. They are in charge of the safety of both adult and child patients while they are hospitalized. The number of patient safety accident reports has rapidly increased, from 563 cases in 2016 to 13,919 cases in 2020 [8].

Indeed, 76.4% of patient safety accident reports submitted through the patient safety reporting system are centred on patient safety coordinators [9]. In Korea, a patient safety coordinator is a “doctor, dentist, oriental medicine doctor, pharmacist, and nurse who has worked at a health care institution for at least 3 years after obtaining the relevant license” [10]. The coordinator is dedicated to tasks related to patient safety and medical quality improvements such as education [10,11], and as of 2016, 99.6% of patient safety coordinators were reported as being assigned to nurses [12].

Small and medium-sized hospitals account for 89.2% of all medical institutions [13], but they are recognized as medical institutions in need of policy support due to the overall weak medical environment [2,14,15]. Although the importance of patient safety is further emphasized in small and medium-sized hospitals [16], it was found that 50% of patient safety coordinators were unable to concentrate on carrying out patient safety activities because of other tasks [17]. Therefore, it should be a priority to determine the degree of role performance so that patient safety coordinators can perform their duties properly in small and medium-sized hospitals [6,18].

To provide a safe medical environment within a medical institution and to strengthen patient safety capabilities, it is important for all medical staff, including hospital management, to work on this together [6]. However, the negative perception that patient safety activities do not directly affect hospital management performance [19], the heavy workload [20], and the unclear standards of the Patient Safety Act may hinder patient safety coordinators from performing their roles adequately [21].

Professional autonomy is one of the factors that influence patient safety management activities in small and medium-sized hospitals [22]. The negative view is that the smaller the hospital is, the less systematic support there is for creating patient safety [23], and the hospital management will prioritize cost reduction or utility over patient safety [24]. It is difficult for patient safety coordinators to make decisions that enable them to perform their roles adequately.

Therefore, in this study, the role of performance and the influence factors of patient safety coordinators in small and medium-sized hospitals are identified, ways to strengthen these hospitals’ patient safety competency are sought, and basic data for improving the patient safety culture are provided.

## 2. Materials and Methods

### 2.1. Study Design

This study is a cross-sectional study to identify the effects of role conflict and professional autonomy on role performance for patient safety coordinators in small and medium-sized hospitals in Korea. During February 2022 (about three weeks), a self-reported questionnaire with 95 items was distributed to the participants via online.

### 2.2. Study Participants

The participants of this study were patient safety coordinators working in hospitals with more than 100–300 beds. Those who have only been registered by submitting an application to the Ministry of Health and Welfare, and those who have been at work for less than one month, were excluded. The sample size was calculated using the G*power 3.1.9.7 program (Düsseldorf, Germany) at the significance level 0.05, effect size 0.15, statistical power 0.80, and 10 predictors. The minimum number of participants required was 118. A total of 132 participants completed the questionnaire, but 11 were excluded due to a lack of standard data. Finally, data from 121 participants were included in the analysis.

### 2.3. Instruments

#### 2.3.1. General and Work-Related Characteristics

In this study, general characteristics and work-related characteristics account for a total of 14 items. The general characteristics of the participants included age, gender, marital status, educational background, job, salary, and motivation for selection. As for work-related characteristics, the type of medical institution, the number of beds, the department of the coordinators, the status of concurrent positions, medical institution evaluation experience, clinical experience, and working experience of coordinators were included.

#### 2.3.2. Role Conflict

Role conflict of the participants was measured with the tool developed for nurses by Kim and Park [25] and further modified for patient safety coordinators. This tool has a total of 26 items, including 12 items on ambiguous roles, 6 items on lack of ability, 5 items on overload, and 3 items on lack of cooperation. Each item was composed of a Likert 5-point scale ranging from “Not at all” to “Always”. The score ranges from a minimum of 26 points to a maximum of 130 points, and the higher the total score, the higher the role conflict. In this study, Cronbach’s α was 0.95.

#### 2.3.3. Professional Autonomy

The professional autonomy of the participants was measured with the tool developed by Dempster [26] and further modified for patient safety coordinators. This tool has a total of 30 items, comprising 11 items on readiness, 7 items on empowerment, 9 items on actualization, and 3 items on evaluation. The score ranges from a minimum of 30 points to a maximum of 150 points, and the higher the total score, the higher the professional autonomy. Each item was composed of a Likert 5-point scale ranging from “Not at all” to “Strongly agree”; items 8, 13, 17, and 28 were reversely scored when calculated, with a total score of all 30 items. In this study, Cronbach’s α was 0.93.

#### 2.3.4. Role Performance

The tool of role performance was developed and used by the authors.

First, a total of 23 preliminary questions on the role of patient safety coordinators were extracted based on the main work elements of the Patient Safety Act Operation Manual.

A total of four members were evaluated: two nursing professors and two supervisor-level clinical experts working in the patient safety department. The Content Validity Index (CVI) of each item was on a 4-point Likert scale: “very appropriate”, 4 points; “content is appropriate but needs partial correction”, 3 points; “necessary content but overall content revision”, 2 points; and “very inappropriate”, 1 point. After reviewing the content validity of the first expert, 4 items with a CVI of 0.75 were revised out of a total of 23 items, and 2 items were added to make a total of 25 secondary items to convey the meaning. As a result of four experts examining the validity of the revised second question in the same way as the first question, all questions scored 0.8 or higher, and the text was modified so that there was no ambiguity.

To determine the correlation between each of the 25 items and the overall role performance tool, the modified item–total correlation coefficient was obtained. As a result of checking the contribution to the tool, all of the correlation coefficients were r > 0.30, so all 25 items were selected without any deleted items. As a result of the exploratory factor analysis, the result of Kaiser–Meyer–Olkin (KMO) was 0.90, and the result of Bartlett’s sphericity test also showed a statistically significant difference (χ^2^ = 2000.73, *p* < 0.001). The items used in the analysis were suitable for conducting a factor analysis. Principal component analysis and varimax rotation were used to maintain the mutually independent relationship between the factors and to identify the characteristics of the factors. As a result, five factors with eigenvalues of 1.0 or higher were extracted, and all 25 items showed a significant factor loading of 0.4 or higher. Resulting from the factor analysis, the five factors explained 67.17% of the total variance in the role performance.

The final developed tool consisted of a total of 25 items. The score ranged from a minimum of 25 points to a maximum of 125 points, and the higher the total score, the better the role performance. In this study, Cronbach’s ⍺ was 0.94.

### 2.4. Data Collection and Human Participants’ Protections

This study was approved by the Institutional Review Board (IRB, PNU IRB/2021_183_HR) of Pusan National University, and data were collected from 8 to 27 February 2022. The online questionnaire was collected via the online platform Moaform (http://www.moaform.com, accessed on 27 February 2022), which could restrict IP access so that participants did not repeat the survey. The internet address (Uniform Resource Locator, URL) or QR code for the online questionnaire was posted on the online community of patient safety coordinators with a document detailing the research purpose, data collection method, and procedure. When people who wanted to participate in the study read this and accessed the online questionnaire, the study purpose and method, standard for participation, and data usage were provided once again on the first screen of the questionnaire and, if they agreed to participate in the survey, they could click the button below to respond, so all participants were voluntarily participating. It was explained that the collected data would be used only for study purposes, that consent to participate in the study could be withdrawn at any time, and that there would be no disadvantages resulting from this. For items that collected personally identifiable information, the contact information was collected to send gift-cons (coffee drink coupons) as a reward for participating in the survey, then immediately disposed of after the reward was completed; only the data necessary to the study were stored on the researcher’s personal computer. Security was maintained by restricting access using passwords.

### 2.5. Data Analysis

The collected data were analyzed with SPSS version 25.0 (SPSS, Inc., Chicago, IL, USA). The statistical significance was set to *p* < 0.05 (two-tailed). To analyze the general and work-related factors of the participants, a descriptive analysis, including frequencies, percentages, means, and standard deviations (SD), was performed. Means and SDs were calculated from the total scores of role conflict, professional autonomy, and role performance. Independent samples *t*-test and one-way ANOVA were applied to analyze the differences in role conflict, professional autonomy, and role performance according to the characteristics of the participants. The post-test was analyzed as the Scheffé test. The correlation between role conflict, professional autonomy, and role performance of participants was analyzed with Pearson correlation coefficients, and a multiple regression analysis was used to identify factors affecting role performance.

## 3. Results

### 3.1. General and Work-Related Characteristics

In terms of the general characteristics of the participants, the average age was 40.62 years, and females accounted for 97.5%. Most participants (60.3%) had a bachelor’s degree, with the most frequent occupation being nursing at 99.2%. In terms of the work-related characteristics of the participants, the average number of beds was 230.84, and 49.6% of the departments were for nursing. It was found that 64.5% of participants performed other tasks in addition to patient safety tasks. Participants who had medical institution accreditation evaluation accounted for 76.9%, followed by medical adequacy evaluation at 47.9% and medical quality evaluation at 24.0%. The average working experience of the participants was 2.34 years (Table 1).

### 3.2. The Degree of Role Conflict, Professional Autonomy, and Role Performance

Table 2 shows the role conflict, professional autonomy, and role performance of the research participants. The average total score for role conflict was 92.50 ± 14.96 out of a total of 130 points. As for the subdomains, role ambiguity scored 44.66 ± 8.09 out of 60 points, and lack of ability was 21.92 ± 3.75 out of 30 points. Overwork was 18.42 ± 3.34 out of 25, and lack of cooperation was 11.26 ± 2.37 out of 15. The average total score for professional autonomy was 95.21 ± 16.67 out of a total of 150 points, and the average total score for role performance was 93.97 ± 14.35 out of a total of 125 points.

### 3.3. Differences in Role Conflict, Professional Autonomy, and Role Performance

Table 3 shows the differences in role conflict, professional autonomy, and role performance according to the characteristics of the participants. For those with a master’s degree, the average role conflict was the highest with 100.83 points (F = 3.51, *p* = 0.033). As a result of the Scheffé test, those with master’s degrees showed that the role conflict was significantly higher than that of those with bachelor’s degrees. Professional autonomy was high when the annual salary was less than KRW 20 million (F = 3.64, *p* = 0.015), and as a result of the Scheffé test, the annual salary of KRW 40 million or more was significantly higher than that of the KRW 20 million to 30 million group. In addition, the group performing concurrently with other tasks had a higher role conflict than the case where only patient safety tasks were performed, which was statistically significant (*t* = 2.06, *p* = 0.041). Lastly, the role performance in the case of medical institution accreditation evaluation (F = 2.23, *p* = 0.028) and the case of medical quality evaluation (F = 2.06, *p* = 0.043) was statistically significantly higher than that with no evaluation.

### 3.4. Correlation between Role Conflict, Professional Autonomy, and Role Performance

Table 4 shows the correlations between the role conflict, professional autonomy, and role performance of participants. Role conflict had a negative correlation with professional autonomy (r = −0.343, *p* < 0.001) and role performance (r = −0.248, *p* = 0.006), while professional autonomy had a positive correlation with role performance (r = 0.312, *p* < 0.001) and was statistically significant.

### 3.5. Factors Affecting Role Performance

Table 5 shows the factors that affected the role performance of the participants. Role conflict and professional autonomy, which were significantly correlated with the participant’s role performance, medical institution certification evaluation, and medical quality evaluation, which were statistically significant in the descriptive statistics, were selected as major variables. A multiple regression analysis was performed using the input method for the variables. Among them, medical institution certification evaluation and medical quality evaluation, which are nominal scales, were analyzed by converting them into dummy variables. In this study, to check whether the regression analysis assumption was satisfied, the Durbin–Watson statistic was 1.895, close to 2, and it was confirmed that there was no autocorrelation. The tolerance was greater than 0.1 from 0.859 to 0.956, and the Variance Inflation Factor (VIF) was less than 10 (from 1.046 to 1.164), indicating that all variables had no multicollinearity problem. As a result of analyzing the factors affecting the role performance of the patient safety coordinators engaged in small and medium-sized hospitals, the regression model was found to be significant (F = 6.988, *p* < 0.001), and the factor most influencing the role performance was professional autonomy (β = 0.279, *p* = 0.002). This was followed by medical institution certification evaluation (β = 0.246, *p* = 0.005) and medical quality evaluation (β = 0.181, *p* = 0.035), and the explanatory power of these factors was 16.6%.

## 4. Discussion

This study attempted to identify factors affecting the role performance of patient safety coordinators in small and medium-sized hospitals, to find ways to enhance their role performance, and to create basic data to improve the patient safety culture of these hospitals.

It was found that 99.2% of the participants were nurses [12], and most of the patient safety coordinators were still nurses. The reason that nurses account for the majority is probably that nurses are recognized as risk managers [27] who create a safe hospital culture and manage risk factors that can affect patient safety. In addition, they are considered to be an expert group that can sensitively identify problems related to patient safety in a complex medical environment and perform active patient safety nursing activities [28]. Recently, the area of patient safety has been emphasized, such as via efforts to integrate patient safety management into the Korean nursing education curriculum [29]. If a legal qualification certification system is prepared through education, it is thought that it will be helpful for the patient safety coordinators to perform their roles.

In this study, it was found that 64.5% of participants performed tasks other than patient safety tasks. This is similar to the results of a previous study [30], in which 62.4% of the total surveyed medical institutions were in charge of patient safety tasks concurrently with other tasks. This is based on the research result that managers’ awareness of patient safety is lower than that of medical staff [12] and that hospital managers tend to reduce the financial costs required for operation [31]. So, the task of patient safety coordinators is indirectly a matter of generating profits, which seems to have affected whether staff consider taking the position. In addition, the results of this study showed that role conflict was higher when the patient safety coordinators performed tasks related to patient safety together with other tasks compared to when they only performed tasks related to patient safety. To reduce the role conflict due to the concurrent role of the patient safety coordinator, the Patient Safety Act currently has legal restrictions on concurrently engaging in other tasks [11], but it is necessary to determine whether these regulations are implemented in the clinical field.

The role conflict of the participants was found to be 3.56 out of 5. Compared with studies using the same tool [32,33], this was higher. Because the role of a patient safety coordinator is different from that of a general nurse, various competencies such as leadership and communication skills as well as professional knowledge are required. In addition, since many tasks need to be resolved by continuously collaborating with the hospital’s medical department, nursing department, administrative department, and management [34], it is thought that role conflict is higher for patient safety coordinators than for other occupations. As a patient safety culture within medical institutions has not yet been established [6,11], it is considered difficult to induce cooperation from other departments to perform tasks related to patient safety. To establish a patient safety culture in medical institutions, making an effort to raise awareness of patient safety in medical workers, such as through research results [6], will be a positive factor in the role performance of patient safety coordinators.

The professional autonomy of the participants was 3.17 out of 5. This was lower than in a previous study [35] that used the same tool. This is thought to be related to the fact that the average career of patient safety coordinators in small and medium-sized hospitals was short due to the Patient Safety Act having been only recently enacted. In addition, the patient safety standards specified in the Patient Safety Act Operation Manual [11] were comprehensively presented, making it difficult to distinguish between tasks and roles, which is thought to have affected the autonomy of professionals. To improve professional autonomy, if the tasks stipulated in the Patient Safety Act are structured more clearly and responsibilities and authority are given accordingly, it is thought that it would be helpful for the patient safety coordinator to figure out the scope of the role.

In this study, the factors affecting the role performance of patient safety coordinators in small and medium hospitals were analyzed by multiple regression analysis. As a result, the biggest factor was professional autonomy. There were few prior studies on the effect of professional autonomy of patient safety coordinators on role performance, so the comparison was difficult. However, professional autonomy was found to be a factor affecting patient safety in Lee’s study of nurses [36], which was consistent with the result from Hwang’s study of nurses in small and medium-sized hospitals [22] that the higher the professional autonomy, the better the performance of patient safety management activities. Through these study results, we see that the higher the professional autonomy of patient safety coordinators, the better their role performance will be. Professional autonomy refers to autonomously making decisions and taking responsibility for them based on the authority gained from professional knowledge and position [37]. Therefore, it is necessary to check whether patient safety coordinators in small and medium-sized hospitals can independently manage patient safety tasks with a focus on professional autonomy.

In this study, we found that the role conflict of the participants did not have a statistically significant effect on the role performance. There was no study targeting patient safety coordinators, so direct comparison was difficult. However, in previous studies [38,39,40], the lower the role performance, the higher the role conflict. In view of this, repeated studies are needed on factors affecting the role performance of the patient safety coordinator.

This study found ways to strengthen the patient safety capabilities of small and medium-sized hospitals and provided a basis for using it as basic data for the development of educational programs to improve the patient safety culture. In addition, the role of the patient safety coordinator is essential to improving patient safety and quality of care, and this study is meaningful in that it provides the basis for preparing effective strategies for practice.

## 5. Conclusions

This study concludes that most of the work of patient safety coordinators is performed exclusively by nurses, so systematic education such as professional nursing courses to enhance competency should be prepared. In addition, legal and institutional role regulations are needed so that the patient safety coordinator can take the lead in creating a patient safety culture, and if the scope of work specified in the Patient Safety Act is more specifically structured, it will help patient safety coordinators to define and perform the role at work. Moreover, it should be considered whether there should be a division of roles between working with children admitted to a hospital and adult patients.

Based on this study, we suggest conducting a future study related to role performance by comparing patient safety coordinators who work in medical institutions with more than 300 beds. In addition, as there is a lack of previous studies targeting patient safety coordinators, a wide range of studies could be conducted to help improve patient safety and quality of care by identifying various variables.

## Figures and Tables

**Table 1 ijerph-19-09392-t001:** General and work-related characteristics (*N* = 121).

Characteristics	Categories	Mean ± SD or *N* (%)
*General characteristics*		
Age (years)		40.62 ± 8.15
Gender	Female	118 (97.5)
	Male	3 (2.5)
Marital state	Married	87 (71.9)
	Unmarried	34 (28.1)
Educational background	Diploma	30 (24.8)
	Bachelor’s degree	73 (60.3)
	Master’s	18 (14.9)
Occupation	Nurse	120 (99.2)
	Doctor	1 (0.8)
Salary (Unit: 10,000 KRW)	<2000	2 (1.7)
	2000–3000	12 (9.9)
	3000–4000	77 (63.6)
	≥4000	30 (24.8)
Motivation for choosing a job	Because of the type of work	50 (41.3)
	Instructions from the hospital	47 (38.8)
	Concern for patient safety	24 (19.8)
*Work-related characteristics*		
Type of medical institution	Convalescent hospital	48 (39.7)
	General hospital	38 (31.4)
	Hospital	23 (19.0)
	Psychiatric hospital	12 (9.9)
The number of beds		230.84 ± 58.90
Affiliation department	Nursing (part)	60 (49.6)
	Solo	35 (28.9)
	Administrative (part)	26 (21.5)
Concurrent position	Work with other tasks	78 (64.5)
	Only perform patient safety tasks	43 (35.5)
Medical institution certification evaluation ^1^	Enforced	93 (76.9)
	Not enforced	28 (23.1)
Adequacy evaluation ^1^	Enforced	58 (47.9)
	Not enforced	63 (52.1)
Medical quality evaluation ^1^	Enforced	29 (24.0)
	Not enforced	92 (76.0)
Total work experience (years)		14.39 ± 8.11
Experience as a dedicated patient safety coordinator (years)		2.34 ± 1.97

^1^ Duplicate response.

**Table 2 ijerph-19-09392-t002:** The degree of role conflict, professional autonomy, and role performance (*N* = 121).

Variables	Mean ± SD	Min	Max
Role conflict	92.50 ± 14.96	36	123
Role ambiguity	44.66 ± 8.09	16	60
Lack of ability	21.92 ± 3.75	11	30
Overwork	18.42 ± 3.34	7	25
Lack of cooperation	11.26 ± 2.37	3	15
Professional autonomy	95.21 ± 16.67	49	137
Role performance	93.97 ± 14.35	32	123

**Table 3 ijerph-19-09392-t003:** Differences in role conflict, professional autonomy, and role performance (*N* = 121).

Variables		Role Conflict		Professional Autonomy		Role Performance	
		Mean ± SD	*t*/F (*p*)	Mean ± SD	*t*/F (*p*)	Mean ± SD	*t*/F (*p*)
Age (years)	<35	92.60 ± 12.24	0.41	92.67 ± 17.57	0.50	91.03 ± 14.06	0.96
	35–40	92.83 ± 13.70		93.74 ± 18.54		94.39 ± 13.70	
	40–45	94.13 ± 16.10		96.63 ± 17.44		93.45 ± 16.30	
	≥45	90.07 ± 17.13		97.07 ± 13.35		97.23 ± 12.34	
Gender	Female	92.25 ± 15.04	−1.12	94.96 ± 16.41	−1.03	93.92 ± 14.51	−0.21
	Male	102.00 ± 7.21		105.00 ± 27.79		95.67 ± 5.51	
Marital state	Married	91.76 ± 15.86	−0.87	95.78 ± 16.09	0.61	94.55 ± 14.49	0.72
	Unmarried	94.36 ± 12.58		93.55 ± 18.51		92.27 ± 14.24	
Educational	Diploma ^a^	91.97 ± 16.93	3.51 *	91.30 ± 14.77	1.59	91.17 ± 15.69	0.95
background	Bachelor ^b^	90.66 ± 14.11	b < c	97.34 ± 16.68		95.36 ± 13.75	
	Master’s ^c^	100.83 ± 12.62		93.06 ± 19.00		93.00 ± 14.47	
Salary	<2000 ^a^	82.00 ± 25.46	1.28	105.50 ± 7.78	3.64 *	91.50 ± 10.61	0.27
(Unit: 10,000)	2000–3000 ^b^	99.50 ± 14.79		84.50 ± 15.20	b < d	91.33 ± 16.78	
	3000–4000 ^c^	92.03 ± 14.64		94.17 ± 17.64		93.84 ± 14.74	
	≥4000 ^d^	91.60 ± 15.15		101.47 ± 12.13		95.50 ± 12.88	
Motivation for	Because of the type of work	90.68 ± 15.43	1.11	96.10 ± 16.71	0.26	93.54 ± 12.85	0.17
choosing a job	Instructions from the hospital	92.53 ± 15.35		93.83 ± 17.96		94.89 ± 13.49	
	Concern for patient safety	96.21 ± 12.96		96.04 ± 14.29		93.04 ± 18.86	
Type of medical	Convalescent hospital	89.00 ± 17.70	1.99	96.83 ± 17.32	0.97	95.38 ± 13.45	1.02
institution	General hospital	95.79 ± 13.65		96.74 ± 17.29		90.71 ± 16.41	
	Hospital	95.65 ± 10.40		92.30 ± 12.44		96.35 ± 12.22	
	Psychiatric hospital	90.00 ± 11.94		89.42 ± 19.01		94.08 ± 14.54	
The number of	100–200	94.35 ± 16.63	1.79	97.87 ± 16.78	0.60	93.57 ± 11.97	0.22
beds	200–250	89.18 ± 13.34		93.31 ± 17.09		95.09 ± 14.14	
	250–300	94.51 ± 15.29		95.66 ± 16.39		93.19 ± 15.61	
Affiliation	Nursing (part)	91.57 ± 13.50	1.82	94.50 ± 15.09	0.54	92.87 ± 12.57	1.36
department	Solo	90.49 ± 17.38		97.63 ± 13.74		97.29 ± 15.75	
	Administrative (part)	97.35 ± 14.18		93.58 ± 22.97		92.04 ± 16.01	
Concurrent	Work with other tasks	94.55 ± 12.88	2.06 *	93.64 ± 15.64	−1.40	92.40 ± 14.85	−1.63
position	Only perform patient safety tasks	88.77 ± 17.70		98.05 ± 18.25		96.81 ± 13.08	
MICE	Enforced	91.54 ± 14.69	−1.29	94.43 ± 16.56	−0.93	95.54 ± 13.69	2.23 *
	Not enforced	95.68 ± 15.69		97.79 ± 17.09		88.75 ± 15.50	
Adequacy	Enforced	90.45 ± 15.13	−1.45	95.22 ± 15.79	0.01	96.19 ± 12.47	1.65
evaluation	Not enforced	94.38 ± 14.67		95.19 ± 17.58		91.92 ± 15.70	
Medical quality	Enforced	91.59 ± 12.50	−0.37	97.48 ± 15.13	0.84	98.00 ± 10.95	2.06 *
evaluation	Not enforced	92.78 ± 15.71		94.49 ± 17.15		92.70 ± 15.09	
Total work	<10	94.14 ± 14.60	0.88	94.03 ± 17.49	0.83	90.42 ± 14.47	1.23
experience	10–15	89.19 ± 13.24		92.15 ± 15.78		94.30 ± 12.21	
(years)	15–20	94.63 ± 12.61		96.06 ± 15.89		95.16 ± 16.49	
	≥20	91.04 ± 19.30		98.96 ± 17.49		97.08 ± 13.14	
Experience of	<1	92.04 ± 15.10	1.39	94.32 ± 17.70	0.70	91.48 ± 18.64	0.34
dedicated	1–3	90.75 ± 15.45		95.89 ± 16.36		94.85 ± 13.85	
The PSC	3–5	97.17 ± 10.95		92.63 ± 17.71		94.20 ± 11.97	
	≥5	89.55 ± 20.24		100.82 ± 13.03		94.55 ± 12.84	

* *p* < 0.05 ^a,b,c,d^ Scheffé test MICE = Medical Institution Certification Evaluation PSC = Patient safety coordinator.

**Table 4 ijerph-19-09392-t004:** Correlation between role conflict, professional autonomy, and role performance (*N* = 121).

Variables	1	1-1	1-2	1-3	1-4	2	3
1 Role conflict	1	0.936 ******	0.881 ******	0.828 ******	0.797 ******	−0.343 ******	−0.248 *****
1–1 Role ambiguity		1	0.749 ******	0.657 ******	0.645 ******	−0.310 *****	−0.249 *****
1–2 Lack of ability			1	0.677 ******	0.641 ******	−0.334 ******	−0.226 *****
1–3 Overwork				1	0.706 ******	−0.276 *****	−0.199 *****
1–4 Lack of cooperation					1	−0.242 *****	−0.121
2 Professional autonomy						1	0.312 ******
3 Role performance							1

** *p* < 0.001; * *p* < 0.05.

**Table 5 ijerph-19-09392-t005:** Factors affecting role performance (*N* = 121).

	B	SE	ß	*t*	*p*	Tolerance	VIF
Professional autonomy	0.240	0.077	0.279	3.116	0.002	0.865	1.157
Medical institution certification evaluation	8.342	2.926	0.246	2.851	0.005	0.931	1.074
Medical quality evaluation	6.074	2.853	0.181	2.129	0.036	0.956	1.046
Role conflict	−0.112	0.086	−0.117	−1.296	0.197	0.859	1.164

F = 6.988, *p* < 0.001, R^2^ = 0.194, Adjusted R^2^ = 0.166, Durbin–Watson = 1.895.

## Data Availability

The datasets generated during and/or analyzed during the current study are available from the corresponding author upon reasonable request.

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
