# Peer review of "The Effect of Role Conflict and Professional Autonomy on the Role Performance of Patient Safety Coordinators in Small and Medium-Sized Hospitals in Korea"

_ijerph, 2022, doi:10.3390/ijerph19159392_

Round 1

Reviewer 1 Report

The manuscript discusses the effects of role conflict and professional autonomy on role performance in patient safety coordinators in small and medium-sized hospitals. The authors clearly showed that systematic education and legal and institutional arrangements for independent roles and work regulations are needed to strengthen patient safety coordinators' competency in small and medium-sized hospitals.

I have only some recommendations

1-     The language, grammar, and sentence structures need to be proofread

2-     More details are required for the study design  

3-     The study type is not a descriptive one

4-     The IRB approval number and the guidelines used in this study did not mention in the methods section

5-     The P symbol should be italic and capitalized   

6-     M letter in the tables should be changed to mean  

7-     More discussion is needed on the factors.

8-     The references style should be uniform 

Author Response

We are grateful for your suggestion and comments. To be more clear and in accordance with the reviewer’s concerns, we have added a brief description and have revised the manuscript. We appreciate your warm work earnestly and hope that the correction will meet with approval.

Reviewer 2 Report

In the Study Subjects section, it would be better to give a number of hospitals with 100-300 beds.

Author Response

We are grateful for your time and comments. To be more clear and in accordance with the reviewer’s concerns, we have added a brief description and have revised the manuscript. We appreciate your warm work earnestly and hope that the correction will meet with approval.

Reviewer 3 Report

- As this paper specifically relates to role conflict and professional autonomy on role performance of patient sfaety in small and medium sized hospitals in Korea, I think that both the title and Abstract need to include "Korea". This is because services, and service delivery in the management of patient safety do vary considerably around the world. By adding the country, the context will become more meaningful for readers.

- The current vocabulary for people participating in research is "participants". Please change "subject/s" to "participant/s" in the script.

- Introduction, page 1, line 33: Do you mean "behind" rather than "later"?

- Introduction, page 1, second paragraph: You give an example of childhood leukaemia , but it is unclear in your paper whether your study refers to adult and paediatric services? You need to develop your literature further to reflect risk management in both adult and paediatric services. Also, do you have paediatric wards in the hospitals, or do the children receive adult nursing care? This needs to be clarified, as many countries have distinct paediatric services and paediatric nurse and AHP practitioners. Distinction between the two in your paper will help the readership understand the main outcomes.

- Page 2, line 46: What is an "oriental doctor"? Do you mean staff who are employed but who have been trained in other countries? No mention is made of Allied Health Professionals, i.e.Speech and Language Therapists, Physiotherapists and OTs. These staff play a MAJOR role in acute hospital care, both within adult and paediatric wards. You may need to comment on therapy provision and the potential benefits in risk reduction in relation to acute care.

- Results, page 4, line 177: Phrase the sentence beginning "Bachelors degree..." to "Most participants (60.3%) had a Bachelors Degree with the most significant occupation being nursing at 99.2%."

Author Response

We appreciate your comments on our manuscript. Those comments are all valuable and very helpful for revising and improving our manuscript. We have reviewed and revised carefully the comments. Revised portion are marked in red in the manuscript. Thanks again for your comments.
